

# Coral geometry and why it matters

Samuel E. Kahng[1,2,3], Eric Odle[4] and Kevin C. Wakeman[2,4]

[1] Oceanography, University of Hawaii, Honolulu, HI, United States of America
[2] Institute for the Advancement of Higher Education, Hokkaido University, Sapporo, Japan
[3] Kikai Institute for Coral Reef Science, Kikai, Japan
[4] Graduate School of Science, Hokkaido University, Sapporo, Japan

## ABSTRACT

Clonal organisms like reef building corals exhibit a wide variety of colony morphologies and geometric shapes which can have many physiological and ecological implications. Colony geometry can dictate the relationship between dimensions of volume, surface area, and length, and their associated growth parameters. For calcifying organisms, there is the added dimension of two distinct components of growth, biomass production and calcification. For reef building coral, basic geometric shapes can be used to model the inherent mathematical relationships between various growth parameters and how colony geometry determines which relationships are size-dependent or size-independent. Coral linear extension rates have traditionally been assumed to be size-independent. However, even with a constant calcification rate, extension rates can vary as a function of colony size by virtue of its geometry. Whether the ratio between mass and surface area remains constant or changes with colony size is the determining factor. For some geometric shapes, the coupling of biomass production (proportional to surface area productivity) and calcification (proportional to volume) can cause one aspect of growth to geometrically constrain the other. The nature of this relationship contributes to a species' life history strategy and has important ecological implications. At one extreme, thin diameter branching corals can maximize growth in surface area and resource acquisition potential, but this geometry requires high biomass production to cover the fast growth in surface area. At the other extreme, growth in large, hemispheroidal corals can be constrained by calcification. These corals grow surface area relatively slowly, thereby retaining a surplus capacity for biomass production which can be allocated towards other anabolic processes. For hemispheroidal corals, the rate of surface area growth rapidly decreases as colony size increases. This ontogenetic relationship underlies the success of microfragmentation used to accelerate restoration of coral cover. However, ontogenetic changes in surface area productivity only applies to certain coral geometries where surface area to volume ratios decrease with colony size.

# INTRODUCTION

The colony morphology of clonal organisms varies widely between and within taxa and has many physiological and ecological implications associated with adaptation to environmental conditions. For reef building corals, colony morphology has been associated with adaption to light, hydrodynamic regime, sedimentation, and subaerial

Corresponding author
Samuel E. Kahng, Kahng@hawaii.edu

exposure (*Jackson, 1979*; *Chappell, 1980*). While many coral species exhibit morphological phenotypic plasticity in response to environmental conditions, their general colony shape is often conserved and can be characteristic within a taxon (*Veron, 2000*; *Todd, 2008*; *Zawada, Dornelas & Madin, 2019*). Coral colony morphology can dictate a coral's internal light regime (*e.g.*, *Anthony, Hoogenboom & Connolly, 2005*; *Kaniewska, Anthony & Hoegh-Guldberg, 2008*; *Kaniewska et al., 2014*), hydrodynamics and mass transfer (*e.g.*, *Hossain & Staples, 2019*; *Hossain & Staples, 2020*), and exposure to peripheral processes (*Meesters, Wesseling & Bak, 1996*). The general geometric shape of a colony can also dictate and constrain the quantitative relationships between various growth parameters (*Barnes, 1973*) which can have additional ecological implications. For corals, these common growth measurements include linear extension rates (length per unit time), areal growth rate (area per unit time), and calcification rate (mass per unit area per unit time) where calcification (G) is related to volume growth rate *via* average density (M = $\rho$V, where M is mass, V is volume, and $\rho$ is avg density) (*Pratchett et al., 2015*).

For all calcifying organisms including reef building corals, there are two distinct components of growth/production: (1) organic or tissue biomass/biovolume and (2) inorganic or calcium carbonate skeleton (although there is a small fractional component of organic matter in coral skeletons) (*DeCarlo, Ren & Farfan, 2018*). Each aspect of growth requires their own set of elemental resources (*e.g.*, nutrients *vs* carbonate alkalinity) which have characteristically different levels of availability depending on the habitat. For organisms like reef building corals (and coralline algae and sclerosponges), the growth in biomass (*i.e.,* production) is proportional to growth in surface area (*Anthony, Connolly & Willis, 2002*) while the growth in skeleton mass (*i.e.,* calcification) is proportional to volume (*via* density). Therefore, biomass production is directly related to surface area growth. In contrast, skeletal mass is roughly proportional to volume (assuming a constant mean density), so a constant calcification rate leads to a constant volume growth rate (per unit surface area). The relationship between these two aspects of growth can be influenced by coral geometry which may cause one aspect of growth to "geometrically" constrain the other which has important implications for life strategies.

For some geometric shapes, the surface area to volume ratio changes with size potentially causing ontogenetic changes in the ratio of biomass production to calcification and the relationship between various growth parameters causing them to size-dependent. One of the most common metrics of coral growth across all morphologies is linear extension rate due in part to the relative ease in which it can be measured and used for comparative purposes (reviewed in *Pratchett et al., 2015*). Linear extensions rates are generally assumed to be characteristics of a species, independent of size (*e.g.*, *Buddemeier & Kinzie, 1976*; *Hughes & Jackson, 1985*; *Kinzie & Sarmiento, 1986*), and often used as a foundation of the census-based approach to calculate calcium carbonate reef budgets (*e.g.*, *Januchowski-Hartley et al., 2017*; *Perry et al., 2018*; *Perry & Alvarez-Filip, 2019*; *Molina-Hernández et al., 2020*; *Cornwall, Diaz-Pulido & Comeau, 2019*). Changes in linear extensions rates are often interpreted as an environmental signal (*Carricart-Ganivet, 2011*). However, even with a constant rate of calcification, linear extension rates can be colony size-dependent due to the dynamics of surface area to volume ratio which is dictated by the geometry of coral
colony (*e.g.*, *Kahng et al., 2023*). Therefore, the relationship between linear extension rate and colony size for various geometries is important to quantify and understand.

Recent reports have investigated the question whether coral growth is isometric by virtue of the colonial modular design or subject to allometric constraints like non-clonal metazoans (*Edmunds, 2006*; *Dornelas et al., 2017*; *Madin et al., 2020*; *Carlot et al., 2022*; *Medellín-Maldonado et al., 2022*). If growth is allometric with an ontogenetic decrease in growth rate, there may be important implications for ecosystem function based on the reduction of age/size distributions of coral communities from mass mortality events during the Anthropocene (*Dietzel et al., 2020*; *Carlot et al., 2021*). While environmental factors in the form of spatial constraints or partial mortality due to stress or senescence obviously influence these growth parameters (*Barnes, 1973*; *Edmunds, 2008*; *Medellín-Maldonado et al., 2022*), the inherent geometric relationships between growth parameters must be clarified to be able to quantitatively reconcile the different metrics of coral growth which are regularly measured using different techniques under a variety of conditions (*Pratchett et al., 2015*).

In this study, four common reef building coral morphologies are modeled using geometric shapes to illustrate the inherent mathematical relationships between various growth parameters and demonstrate how colony geometry can characteristically alter whether relationships are size-dependent or size-independent. Using previously published empirical values, the geometry-dependent relationships between organic (biomass production) and inorganic (calcification) growth are investigated, and the associated ecological implications are discussed.

## MATERIALS AND METHODS

To demonstrate the inherent mathematical relationships between various coral growth parameters, four geometric shapes were used to model how calcification rate per unit area (G) affects linear extension rate (C) across time, change in surface area ($\Delta S$), change in planar area ($\Delta P$), and change in volume ($\Delta V$): hemispheroids across the full range of eccentricity values (oblate, circular, prolate), flat discs with constant thickness, branching corals with constant branch diameters (cylinder model), and a single conical branch/colony with constant aspect ratio ($\alpha$ = height/basal radius). The accuracy of using simple geometric models has been explicitly tested and confirmed for several diverse coral taxa using 3D scanning and photogrammetry techniques (*Courtney et al., 2007*; *Naumann et al., 2009*). Surface area productivity (SAP = $\Delta S$/S) and planar area productivity (PAP = $\Delta P$/S) were calculated to demonstrate how quickly a coral generates area (per unit area) across time.

Size dependent relationships were illustrated by calculating parameters as a function of size (*e.g.*, radius, thickness, length, or height). Average or common empirical values for G and density ($\rho$) were used from the literature for the various species modeled. To illustrate the effects of geometry independent of G and $\rho$ values, SAP was calculated for all shapes using the same values for G and $\rho$. For this modeling exercise, favorable growth conditions were assumed with no partial mortality or spatial constraints on growth.

A hemispheroid shape was used to model the relationships between these parameters for massive corals (*e.g.*, *Porites* spp.). The effect of the eccentricity (e) was calculated

across the full range of possible values from oblate (flattened) to circular to prolate (tall) hemispheroids. For a circular hemisphere ($e = 0$), the calcification rate $\left(G = \frac{\Delta V \rho}{S_t}\right)$ and volume $\left(V = \frac{2}{3}\pi r^3\right)$ equations were used to solve for extension rate ($C_t = r_{t+1} - r_t$). To calculate SAP $\left(\frac{\Delta S}{S_t} = \frac{S_{t+1} - S_t}{S_t}\right)$, the calcification equations was solved for $r_{t+1}$ and substituted into the equation where surface area $S_t = 2\pi r^2$. PAP $\left(\frac{\Delta P}{S_t} = \frac{P_{t+1} - P_t}{S_t}\right)$ was solved using the equation for planar area $\left(P = \pi a^2\right)$.

For a prolate hemispheroid with a circular base (a=b), the polar radius (c>a), where c is the polar radius and a and b are the equatorial radii in the two horizontal dimensions. The surface area $\left(S = \pi a^2 + \frac{\pi a c}{e}\sin^{-1}e\right)$ and volume $\left(V = \frac{2}{3}\pi a^2 c\right)$ equations were used to solve for SAP using the definition for eccentricity $\left(e = \sqrt{1 - \frac{a^2}{c^2}}\right)$, where c>a. PAP $\left(\frac{\Delta P}{S_1}\right)$ was solved using the equation for planar area $\left(P = \pi a^2\right)$. For an oblate hemispheroid with a circular base (a =b), the polar radius (c<a), where c is the polar radius and a and b are the equatorial radii in the two horizontal dimensions. The surface area $\left(S = \pi a^2 + \frac{\pi c^2}{2e}\ln\left(\frac{1+e}{1-e}\right)\right)$ and volume $\left(V = \frac{2}{3}\pi a^2 c\right)$ equations were used to solve for SAP using the definition for eccentricity $\left(e = \sqrt{1 - \frac{c^2}{a^2}}\right)$, where c<a. PAP $\left(\frac{\Delta P}{S_t}\right)$ was solved using the equation for planar area $\left(P = \pi a^2\right)$. For graphing purposes, oblate eccentricity was denoted as negative values to distinguish it from prolate eccentricity. The effects of eccentricity and size (equatorial radius) on SAP and PAP were calculated.

A flat circular disk with radius (r) and height/thickness (h) was used to model plate-like corals such as those common in the lower photic zone (*i.e., Leptoseris* spp. and *Agaricia* spp.) (*Kahng et al., 2019*; *Tamir et al., 2019*; *Gijsbers et al., 2022*). The calcification rate $\left(G = \frac{\Delta V \rho}{S_t}\right)$ and volume ($V = h\pi r^2$) equations were used to solve for extension rate ($C_t = r_{t+1} - r_t$). To calculate surface area productivity or SAP $\left(\frac{\Delta S}{S_t} = \frac{S_{t+1} - S_t}{S_t}\right)$, the calcification equations was solved for $r_{t+1}$ and substituted into the equation where surface area $S = \pi r^2$. The effect of skeletal thickness (h) on SAP was calculated within the range of empirical values reported (*Kahng et al., 2020*).

A cylinder was used to model branching corals with constant average branch radius (r) (*e.g.,* some branching *Acropora* spp.). As the colony grows, the sum of all branch lengths or total branch length ($h_t$) can be used to calculate skeletal volume, independent of the number of branches ($\beta_t$) or average branch length ($h_t / \beta_t$). The surface area ($S_t = 2\pi r h_t + \pi r^2$) can be calculated from the side of the cylinder using total branch length. Since each new branch base covers the side of its parent branch, there is no net increase in surface area from each new branch tip, with the exception of the area of the very first branch tip ($\pi r^2$). In this model, the ratios between growth in volume ($\Delta V = \pi r^2[h_{t+1} - h_t]$), surface area ($\Delta S = 2\pi r[h_{t+1} - h_t]$), and total linear extension rate ($C_t = [h_{t+1} - h_t]$) remain constant which is consistent with empirical growth data for *Acropora cervicornis* (*Million et al., 2021*). The calcification rate $\left(G = \frac{\Delta V \rho}{S_t}\right)$ and volume ($V = h\pi r^2$) equations were used to solve for average branch extension rate $\left(\frac{C_t}{\beta_t}\right)$, where $C_t = h_{t+1} - h_t$. To calculate surface

**Table 1 Changes in various parameters as coral colony grows in size for each geometric shape.** Vertical arrows denote increase/decrease while slating arrows indicate asymptotic increase/decrease. Equal sign denotes constant values which do not change with colony size.

| | Size metric | Surface area to volume ratio | Linear extension rate | Surface area productivity |
|---|---|---|---|---|
| **Hemispheroid (constant eccentricity)** | equatorial radius | ↘ | ↗ | ↘ |
| **Disc (constant thickness)** | radius | = | ↑ | = |
| **Cone (constant aspect ratio)** | height/length, basal radius | ↘ | ↗ | ↘ |
| **Branching coral - cylindrical (constant branch diameter)** | average branch length | = | ↘ | = |

area productivity or SAP $\left(\frac{\Delta S}{S_t} = \frac{S_{t+1} - S_t}{S_t}\right)$, the calcification equations was solved for $h_{t+1}$ and substituted into the equation where surface area. The effect of branch diameter (2r) on SAP was calculated within the range of empirical values reported in the literature.

A cone with a constant aspect ratio was used to model a single branch of branching corals with conical branches. Some branching corals have branches that mimic high aspect cones. This model also applies to some massive corals form colonies that can mimic low aspect cones. A constant aspect ratio ($\alpha = h_t/r_t$) during growth was assumed, where h is branch length/height and r is radius. The calcification rate $\left(G = \frac{\Delta V \rho}{S_t}\right)$ and volume $\left(V = \frac{1}{3} h \pi r^2 = \frac{\pi h^3}{3\alpha^2}\right)$ equations were used to solve for extension rate ($C_t = h_{t+1} - h_t$). To calculate surface area productivity or SAP $\left(\frac{\Delta S}{S_t} = \frac{S_{t+1} - S_t}{S_t}\right)$, the calcification equations was solved for $h_{t+1}$ and substituted into the equation where surface area. The effects of aspect ratio and cone length/height on SAP was calculated.

## RESULTS

The derivations of the mathematical relationships for each geometric shape are illustrated in Appendix A. Given a constant calcification rate (G) and density ($\rho$), the size-dependence of the linear extension rate ($C_t$) for the four geometric shapes depends on whether the surface area to volume ratio remains constant or changes as the coral colony grows in size (Table 1). For a circular hemispheroid, the size-dependent relationship between calcification rate and (radial) linear extension rate can be expressed $G = \frac{\Delta V \rho}{S_t} = \frac{\rho(3C_t r_1^2 + 3C_t^2 r_t + C_t^3)}{3r_t^2}$, so as the colony reaches a larger size ($r_t \to \infty$), $G \to \rho C_t$. Therefore, the linear extension rate increases to an asymptotic maximum value of $C_t \to \frac{G}{\rho}$ (Fig. 1). For hemispheroids with nonzero eccentricity, the relationships quantitatively depart from the circular hemisphere (equatorial radial extension rate slower for prolate and faster for oblate) but the asymptotic pattern is qualitatively the same.

For a circular flat disc with constant thickness (h), the linear extension rate increases linearly with colony size (radius), $C_t = r_t \left(\sqrt{\frac{G}{\rho h} + 1} - 1\right)$ (Fig. 1) and exponentially across time (*Kahng et al., 2023*).

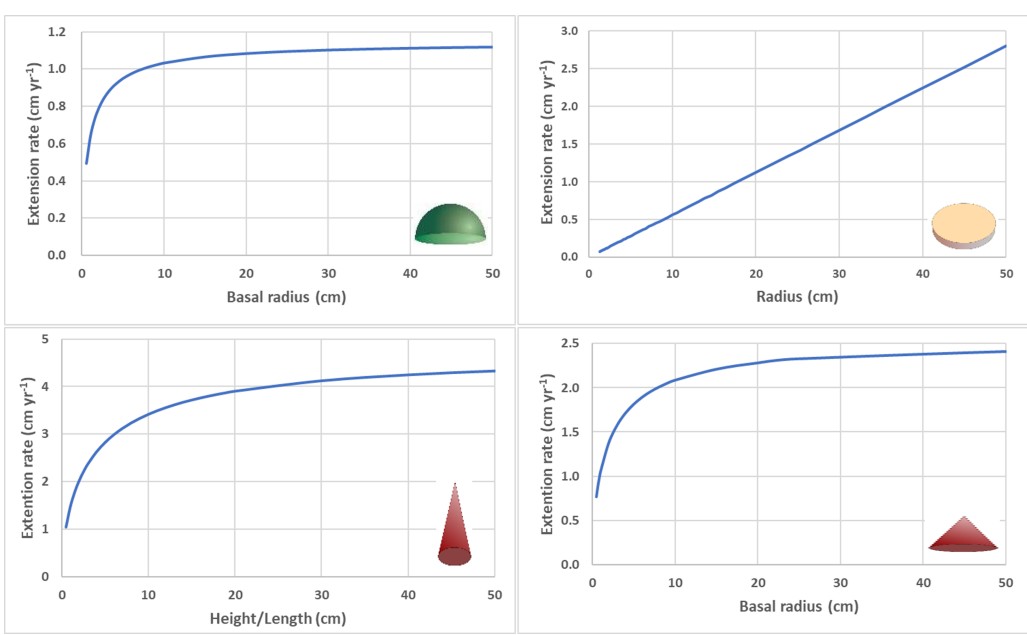

**Figure 1** **Size dependent linear extension rates.** (A) Radial extension rate for circular hemisphere (eccentricity = 0) with constant calcification rate ($G = 1.6$ g cm$^{-1}$ yr$^{-1}$) and density ($\rho = 1.4$ g cm$^{-3}$). (B) Radial extension rate for circular disc with constant calcification rate ($G = 0.06$ g cm$^{-1}$ yr$^{-1}$), density ($\rho = 2.6$ g cm$^{-3}$), and thickness (0.2 cm). (C) Height/length extension rate for cone with high aspect ratio ($\alpha$ = height/radius = 4) with constant calcification rate ($G = 1.6$ g cm$^{-1}$ yr$^{-1}$) and density ($\rho = 1.4$ g cm$^{-3}$). (D) Radial extension rate for cone with low aspect ratio ($\alpha = 0.5$) with constant calcification rate ($G = 1.6$ g cm$^{-1}$ yr$^{-1}$) and density ($\rho = 1.4$ g cm$^{-3}$).

For a cone with constant aspect ratio, the linear extension rate for height increases asymptotically, $C_t = \sqrt[3]{\frac{3G\sqrt{\alpha^2+1}}{\rho}h_t^2 + h_t^3} - h_t$ , where $C_t = h_{t+1} - h_t$. For high aspect cones (*e.g.*, $\alpha > 4$), the linear extension rate will not approach the asymptotic maximum value $\left(C_t \to \frac{G\sqrt{\alpha^2+1}}{\rho}\right)$ until branch length/height become unrealistically long (*e.g.*, h>50 cm) (Fig. 1). The linear extension rate for basal radius also increases asymptotically, $C_t = \sqrt[3]{\frac{3G\sqrt{\alpha^2+1}}{\rho\alpha}r_t^2 + r_t^3} - r_t$ , where $C_t = r_{t+1} - r_t$ . For low aspect cones (*e.g.*, $\alpha < 0.5$), the linear extension rate will approach the asymptotic maximum value $\left(C_t \to \frac{G\sqrt{\alpha^2+1}}{\rho\alpha}\right)$ at relatively low values (*e.g.*, h>25 cm) (Fig. 1).

For a branching coral with cylindrical branches of constant diameter, the relationship between calcification rate and total linear extension rate for all branches can be expressed as $C_t = \frac{2Gh_t}{\rho r} + \frac{G}{\rho}$ , where h$_t$ is the total lengths of all branches combined. Dividing both sides by the number of branches $\beta_t$ provides the average branch extension rate $\frac{C_t}{\beta_t} = \frac{2G}{\rho r}\frac{h_t}{\beta_t} + \frac{G}{\rho\beta_t}$ , where $\frac{C_t}{\beta_t}$ is average branch extension rate and $\frac{h_t}{\beta_t}$ is average branch length. As the colony grows more branches (*i.e.*, $\rho\beta_t >> G$), the average branch extension rate decreases asymptotically to a value proportional to average branch length, $\frac{C_t}{\beta_t} \to \frac{2G}{\rho r}\frac{h_t}{\beta_t}$ . The proportionality constant decreases within increasing branch diameter.

Given a constant calcification rate (G) and density ($\rho$), the size-dependence of the surface area productivity (SAP) also depends on whether the surface area to volume ratio

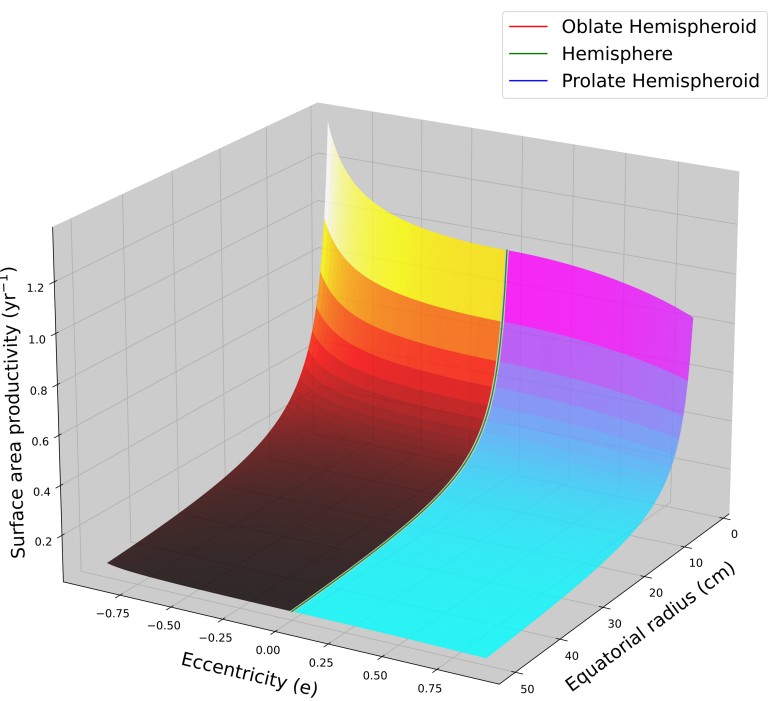

**Figure 2 Hemispheroid surface area productivity.** Surface area productivity ($\mathrm{yr}^{-1}$) for a hemispheroid coral colony as a function of colony size (equatorial radius, a) and eccentricity (e) calculated using average values for massive *Porites*: $G = 1.64\ \mathrm{g\ cm^{-2}\ yr^{-1}}$, $\rho = 1.28\ \mathrm{g\ cm^{-3}}$ (*Lough, 2008*). For graphing continuity purposes, eccentricity values for oblate hemispheroids are denoted as negative.

remains constant or changes as the coral colony grows in size (Table 1). For hemispheroids with constant eccentricity (e), the equations for SAP are listed below:

- Circular hemisphere $\dfrac{\Delta S}{S_t} = \dfrac{\left(\frac{3Gr_t^2}{\rho}+r_t^3\right)^{2/3}}{r_t^2} - 1$

- Prolate hemispheroid $\dfrac{\Delta S}{S_t} = \dfrac{\left(\left(a_t^3 + \frac{a_t^2 3G\sqrt{(1-e^2)}}{2\rho}\left(1+\frac{\sin^{-1}e}{e\sqrt{(1-e^2)}}\right)\right)\right)^{2/3}}{a_t^2} - 1$

- Oblate hemispheroid $\dfrac{\Delta S}{S_t} = a_t^{-2}\left(a_t^3 + \dfrac{3Ga_t^2\left(1+\frac{(1-e^2)}{2e}\ln\left(\frac{1+e}{1-e}\right)\right)}{2\rho\sqrt{1-e^2}}\right)^{2/3} - 1$

Using average G and $\rho$ values for massive *Porites* (*Lough & Barnes, 2000*; *Lough, 2008*; *Lough & Cantin, 2014*; *Lough et al., 2016*) were used to calculated SAP as a function of colony size (equatorial radius) and eccentricity (Fig. 2). Flatter hemispheroids (*i.e.,* larger c/a ratio) have higher SAP than taller ones. For all hemispheroids, SAP quickly declines asymptotically with increasing size as a coral colony grows.

For a cone with constant aspect ratio, SAP is a size dependent and decreases asymptotically with increasing size in a manner analogous to hemispheroids, $\frac{\Delta S}{S_t} =$

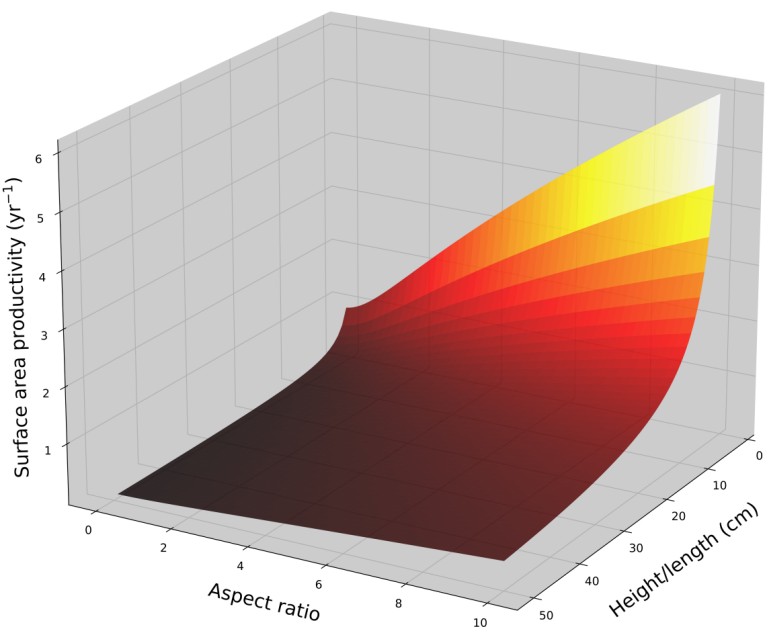

**Figure 3 Cone surface area productivity.** Surface area productivity (yr$^{-1}$) for a single conical coral colony as a function of height (or a single conical branch) and aspect ratio ($\alpha$ = length/basal radius) using average values for branching *Acropora*: $G = 2.5$ g cm$^{-2}$ yr$^{-1}$, $\rho = 1.4$ g cm$^{-3}$ (*Hughes, 1987*; *Morgan & Kench, 2012*).

$\left( \dfrac{3G\alpha\sqrt{1+\frac{1}{\alpha^2}}}{\rho h_t} + 1 \right)^{2/3} - 1$. Using average G and $\rho$ values for branching *Acropora* (*Hughes, 1987*; *Morgan & Kench, 2012*; *Pratchett et al., 2015*), the SAP for a single conical branch was calculated as a function of branch length and aspect ratio (Fig. 3). Aspect ratio is positively correlated with SAP.

For a branching coral with cylindrical branches, surface area to volume ratio remains constant and all growth parameters are linearly related to total branch length (assuming a constant density). SAP is a size-independent and a function of calcification rate and branch radius/diameter (r), $\frac{\Delta S}{S_t} = \frac{2G}{\rho r}$ . SAP declines with increasing branch thickness. Using average G and $\rho$ values for branching *Acropora* spp. (*Hughes, 1987*; *Morgan & Kench, 2012*; *Pratchett et al., 2015*), the SAP was calculated as a function of branch thickness (Fig. 4).

For a flat circular disc, SAP is a size independent and is a function of calcification rate and thickness (h), $\frac{\Delta S}{S_t} = \frac{G}{\rho h}$ . Thinner colonies have a higher SAP. Using average G and $\rho$ values for deep-water *Leptoseris* spp. from 70–111 m (*Kahng et al., 2023*), the SAP was calculated as a function of average skeletal thickness (Fig. 5).

To isolate impact of geometry alone, SAP was calculated using equivalent values for G and $\rho$ for the contrasting shapes (Fig. 6). Flat, thin plate-like shapes have the highest SAP which is equivalent to planar area productivity (PAP) by virtue of their horizontal orientation. Cylindrical branching corals with small branch diameters also exhibit high SAP, especially in comparison to their massive coral counterparts (hemispheroids and low aspect cones) which have very low SAP as they reach larger size.

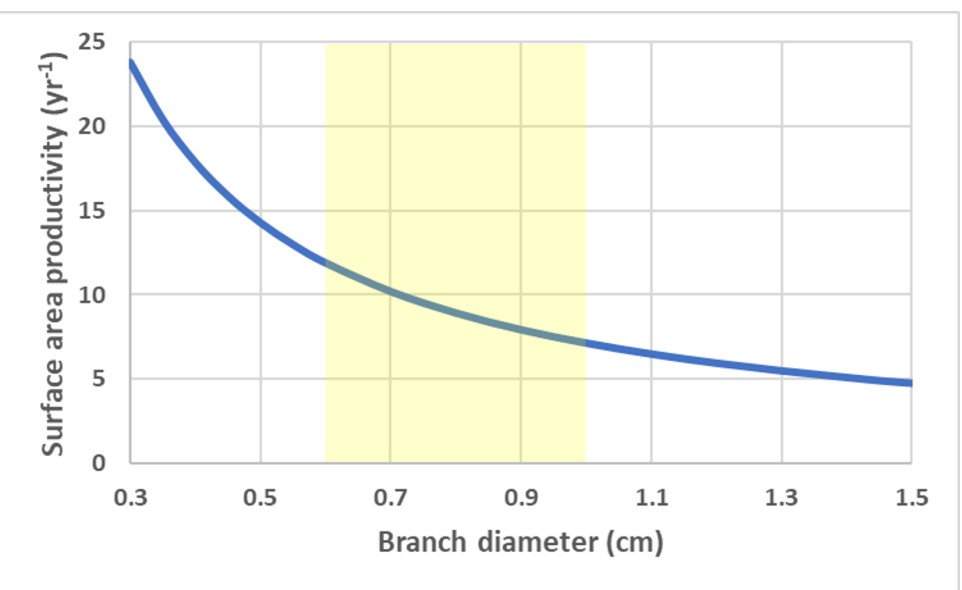

**Figure 4  Cylinder surface area productivity.** Surface area productivity ($yr^{-1}$) for a branching coral colony with cylindrical branches as a function of branch diameter using average values for branching *Acropora*: $G = 2.5$ g cm$^{-2}$ yr$^{-1}$, $\rho = 1.4$ g cm$^{-3}$ (*Hughes, 1987*; *Morgan & Kench, 2012*). The yellow shading indicates common values for branch diameter (*Nadler et al., 2014*).

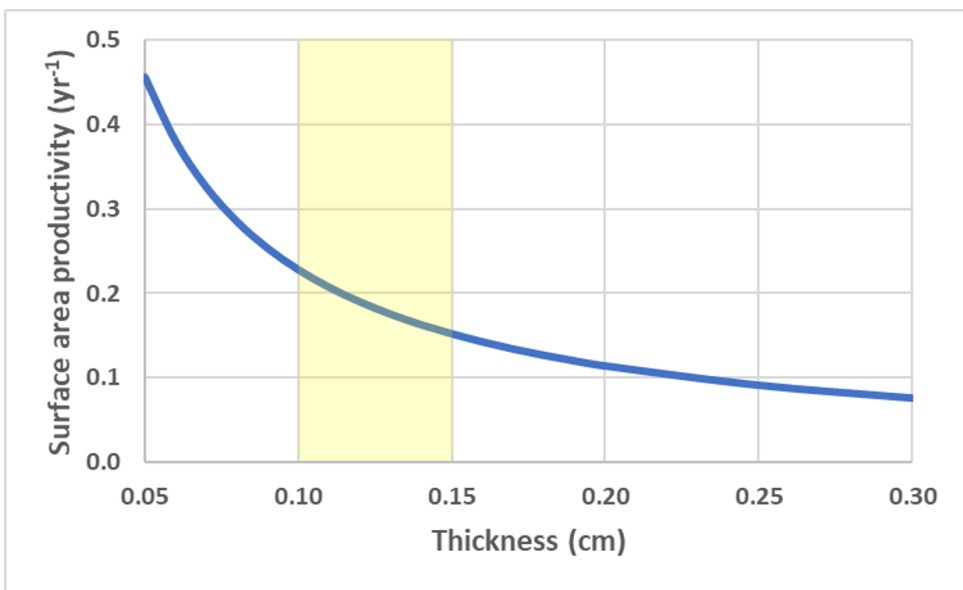

**Figure 5  Disc surface area productivity.** Surface area productivity for a flat plate-like colony as a function of average thickness using average values for branching deep-water *Leptoseris* spp. from 70–111 m in Hawaii: $G = 0.061$ g cm$^{-2}$ yr$^{-1}$, $\rho = 2.67$ g cm$^{-3}$ (*Kahng et al., 2023*). The yellow shading indicates common values for skeletal thickness diameter (*Kahng et al., 2020*).

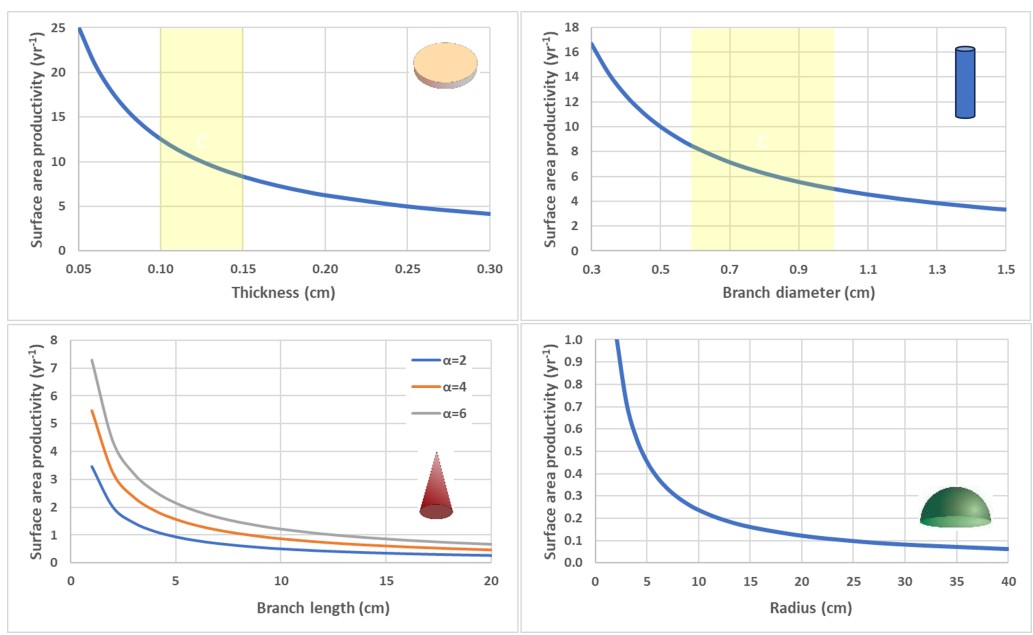

**Figure 6** **Surface area productivity (yr$^{-1}$) for four geometric shapes using equivalent values for calcification rate ($G = 1.5$ g cm$^{-2}$ yr$^{-1}$) and density ($\rho = 1.2$ g cm$^{-3}$).** Flat disc (top left), Cylindrical branching (top right), Single conical branch with a range of aspect ratios ($\alpha$) (bottom right), and Circular hemispheroid (bottom right). Yellow shaded areas highlight common values for deep-water *Leptoseris* spp. skeletal thickness (top left) and *Acropora* spp. branch diameter (top right).

## DISCUSSION

### Linear extension rates

Depending on coral colony morphology, inherent mathematical relationships determine which geometries can be expected to have size-independent or size-dependent linear extension rates (C$_t$) . In general, the stability of the quantitative relationship between calcification rate and linear extension rate depends on the surface area to volume ratio as the colony grows (assuming constant density). Given a constant calcification rate, linear extension rate is expected to be nearly constant for large hemispheroid coral colonies (*e.g.*, radius >25 cm) and large branching coral colonies (*e.g.*, >20 branches) with constant branch thickness and average branch length. However, when these colonies are small, decreases in the surface area to volume ratios are significant, so both radial extension rates of hemispheroidal corals and average branch extension rate for branching corals can be expected to decrease significantly during initial growth (if calcification rate remains constant). Given an equivalent calcification rate and branch diameter, longer average branch lengths cause faster average branch linear extension rates. Similarly, thicker branch diameters cause slower average branch linear extension rates.

For conical shaped colonies with a constant calcification rate and aspect ratio ($\alpha$), linear extension rates for both the height/length and the basal radius will decrease asymptotically with increasing height/length or radius respectively. For high aspect cones (*e.g.*, $\alpha$>4) like those of some branching corals, the natural lengths of conical branches are short relative

to the values needed to approach the maximum extension rates. For low aspect cones (*e.g.,* $\alpha < 0.5$) like some flattened morphologies of massive corals, the basal radius extension rates will approach the maximum values quickly relative to the size and longevity of these corals.

For disc-like colonies with constant calcification rate and mean thickness (h), radial extension rate can be expected to increase throughout the life of the colony linearly with colony size (radius) and exponentially across time (*Kahng et al., 2023*). As the colony grows the ratio of surface area (*i.e.,* resource acquisition potential) to circumference (i.e, the actively growing portion of the colony) increases linearly with size fueling its radial/linear extension rate.

While the accuracy of simple geometric models for several diverse coral taxa has been explicitly tested and confirmed *via* 3D photogrammetry and 3D computer tomography (CT) scans (*Courtney et al., 2007*; *Naumann et al., 2009*), the actual morphology of coral taxa may vary from simple geometric shapes (*e.g.,* *Darke, 1991*; *Darke & Barnes, 1993*). For example, some massive *Porites* increase their bumpiness ontogenetically to increase their surface area to volume ratio (*Darke, 1991*) causing deviations from a smooth hemispheroid model but retaining a characteristic log-linear relationship (*Courtney et al., 2007*). For coral taxa which undergo ontogenetic shifts in basic morphology (*e.g.,* *Montastrea annularis, Porites sillimaniani. etc.*), simple geometric models cannot be applied (*e.g.,* *Graus & Macintyre, 1982*; *Muko et al., 2000*). For corals with intraspecific morphological variations due to habitat specific environmental adaptations (reviewed in *Todd, 2008*), geometric shape parameters (*e.g.,* eccentricity, aspect ratio, thickness, *etc.*) must be appropriately adjusted for each colony and location.

The key factor which determines whether linear extension rates have a size-dependent geometry is whether the ratio of mass ($\rho V$) to surface area changes or remains constant as the colony grows and therefore, whether average density ($\rho$) also remains constant. For massive *Porites* spp., skeletal density varies seasonally on a regular basis, but average density across longer timescales is more stable (*Lough & Barnes, 1990*; *Lough & Barnes, 1992*). A small ontogenetic decrease in density (*i.e.,* 4% per 100 yrs) has been reported (*Lough, 2008*). Within a colony, variations in skeletal density across time are generally less than 10% of mean values and several times lower than the percentage variation in calcification and extension rates (*Lough & Barnes, 2000*; *Razak et al., 2019*). While calcification rate and extension rate correlate with each other, calcification rate does not correlate with density (*Lough & Barnes, 2000*; *Lough et al., 2016*). For coral taxa that do not conform to a simple geometric shape, 3D scanning and photogrammetry (*e.g.,* *Courtney et al., 2007*; *Naumann et al., 2009*; *Zawada, Dornelas & Madin, 2019*; *Million et al., 2021*) can be used to measure surface area and determine whether the surface area to mass ratio (or surface area to volume ratio) is size-dependent. For some branching coral taxa, density varies ontogenetically due in part to secondary infilling (*Hughes, 1987*; *Li et al., 2021*) so confirmation of the surface area to mass ratios and the ratio of their respective growth parameters should be confirmed empirically.

Because calcification rates for a given coral colony can vary considerably across time, the ontogenetic, size-dependent geometric effects on linear extension rate can be
overshadowed. Significant interannual variability (*e.g.*, ±25–50% of long-term mean) is commonly reported from retrospective analyses of massive corals (*e.g.*, *Bolouki Kourandeh et al., 2018*; *Razak et al., 2019*; *Courtney, Kindeberg & Andersson, 2020*; *etc.*). These temporal changes in calcification rate are associated with a variety of biotic and abiotic environmental factors. For hemispheroids and conical massive corals, the asymptotic maximum linear extension rates are directly proportional to calcification rate. For a flat circular disc, linear extension rates are directly proportional to size (*i.e.,* radius) and to (roughly) the square root of calcification rate, so size can have an exponentially larger impact.

Whether a particular growth metric is naturally isometric or allometric depends on both the individual metric and the colony geometry. Of course, the environmental history of each coral has a dominant effect on *in situ* net growth generating wide variability between conspecifics across time (*e.g.*, *Madin et al., 2020*). Nonetheless, the character (whether isometric/allometric) of one growth metric can be characteristically different than another due to their mathematical relationship which is dictated by colony geometry. Likewise, the same growth metric may differ characteristically between corals with dissimilar geometries.

## Organic and inorganic production

For all calcifying organisms, both biomass production and calcification are required for growth. While both require energy, each aspect of growth has different elemental resource requirements, and the availability of these elemental resources can vary significantly by habitat and depth. In oligotrophic waters associated with many (but not all) shallow coral reef ecosystems, the limiting elemental resources for coral biomass production are typically macronutrients like bioavailable nitrogen or soluble phosphorus (*Atkinson & Falter, 2003*; *Karl & Church, 2017*). This limitation of essential macronutrients often results in a surplus and subsequent excretion of organic carbon production (*i.e.,* photosynthate) in well-lit habitats (reviewed in *Goldberg, 2018*). In contrast, the elemental resources for calcification (calcium ions and carbonate alkalinity) are generally not limiting in surface seawater (at normal pH) in tropical habitats. In clear, oligotrophic waters, the depth of the lower photic zone can coincide with the nutricline (or interaction with the nutricline *via* inertial oscillations) causing inorganic macronutrients to be less limiting or even replete (*Kahng et al., 2019*). However, available light energy to drive photosynthesis eventually becomes limiting with increasing depth.

For reef building corals, coralline algae, and sclerosponges, tissue biomass is limited to a finite layer associated with surface area of their skeleton. This design generates a geometric relationship and potential constraint between the two aspects of growth (skeletal mass and tissue biomass). Assuming a constant mean tissue thickness and composition, the tissue biomass can be expected to be proportional to surface area unless there is significant nonliving surface area (due to senescence or partial mortality). Therefore, the surface area productivity (growth of new surface area per unit surface area per unit time) can be a useful metric that can be used as a quantitative proxy for ontogenetic changes in biomass productivity (as the colony grows in size) and how geometry can constrain it. To the extent that tissue thickness/composition changes ontogenetically with size (*e.g.*, *Darke, 1991*;

*Barnes & Lough, 1992*; *Lough & Barnes, 2000*), SAP as a proxy for biomass productivity must be adjusted accordingly.

For calcifying organisms with an external skeleton (*i.e.*, shelled molluscs), calcification can directly relate to the growth of internal volume available for biomass (*e.g.*, *Raup & Graus, 1972*; *Graus, 1974*) generating the potential for an analogous geometric constraint. However, ocean acidification studies have demonstrated that reduced calcification can occur without reductions in somatic growth due to reductions in shell thickness and structural integrity (reviewed in *Gazeau et al., 2013*). Phenotypic plasticity in shell thickness relative to tissue biomass (reviewed in *Watson et al., 2012*) further suggests that despite the potential geometric relationship between organic and inorganic production, the two aspects of growth are readily decoupled.

## Ecological implications of SAP

For clonal organisms with indeterminate growth, rates of resource acquisition (*i.e.,* energy and elemental resources) can limit rates of growth. Within a stable environment in the absence of stress, space limitations, and disturbance, coral resource acquisition can be expected to scale with tissue surface area exposed to light for autotrophy and ambient water for heterotrophy (whether *via* polyp feeding or direct assimilation) and mass transfer (*Monismith, 2007*). Therefore, under optimal conditions maximum rates of production normalized to surface area (*i.e.,* calcification and growth in biomass) can be expected to be independent of colony size. However, depending on colony morphology, portions of a colony's surface area may be subject to self-shading and/or reduced water motion (*e.g.*, feeding and mass transfer) causing reductions, spatial heterogeneity in productivity per unit area, and spatial patterns of polyp senescence/mortality (*e.g.*, *McFadden, 1986*; *Kim & Lasker, 1998*; *Medellín-Maldonado et al., 2022*).

Corals with high SAP can increase their area and therefore resource acquisition capacity quickly to fuel fast growth (both tissue biomass and calcification) when resources are not limiting. However, their growth requires high biomass production to cover the quickly expanding surface area. Therefore, biomass production may limit growth more than calcification with the latter being constrained by the former. This potential constraint has implications for tissue thickness (or biomass reserves) and resilience. The ability of branching corals, especially with species thin branch diameters, to recover from tissue damage or loss can be expected to be less robust (without impacting growth) than corals with lower SAP. Many branching corals are known to have relative thin tissue thickness (*e.g.*, *Loya et al., 2001*). These geometry-based predictions are consistent with the life history traits of fast-growing branching corals (*e.g.*, *Acropora* spp.) and implies that their susceptibility to disturbance such as bleaching-related mortality may be due in part to the inherent tradeoff associated with their geometry (*Loya et al., 2001*; *McClanahan et al., 2007*; *Darling et al., 2012*). Where studied, some corals exhibit an inverse correlation between tissue layer thickness and calcification rate which is consistent with a geometric tradeoff (*Lough et al., 2016*). For corals which characteristically have tissue recession (*i.e.,* senescence) basal to the direction of growing branches/ramets, any biomass reallocation would reduce the demand for new biomass production accordingly.

Several coral taxa (*e.g.*, massive *Porites* spp., *etc.*) have colonies that mimic a hemispheroidal or a low aspect conical shape. Large colonies with these shapes have very low SAP and grow surface area very slowly. Their biomass growth may be geometrically constrained by calcification thereby giving them a surplus in biomass production capacity in excess of growth requirements (*Anthony, Connolly & Willis, 2002*). This slower biomass growth strategy may be tied to their thicker tissues (*Loya et al., 2001*) and be associated with a geometric tradeoff for greater resilience to bleaching (*e.g.*, *Loya et al., 2001*; *Hughes et al., 2018*), a greater capacity to recovery from tissue damage/loss, and higher reproductive output per unit area (*Alvarez-Noriega et al., 2016*). For these corals, slow growth in surface area translates into slow growth in resource acquisition potential which would be particularly disadvantageous during early growth due to size-dependent mortality (*Hughes & Connell, 1987*). However, their S:V ratio changes ontogenetically and are an order of magnitude higher during initial growth and become low only after reaching moderate size. During initial growth, their high SAP (and lack of surplus in biomass production capacity) may contribute to the increased mortality rates of juveniles (*Meesters, Wesseling & Bak, 1997*).

The rapid, initial, ontogenetic change in SAP for massive corals (hemispheroidal or conical) underlies the success of microframentation as a technique for accelerating their growth in coral cover (*Forsman et al., 2015*; *Page, Muller & Vaughan, 2018*). When large, massive corals are cut into small fragments, SAP is fundamentally increased by virtue of their geometry (independent of any resource allocation to reproduction). Given an equivalent calcification rate, small colonies grow surface area multiple time faster (per unit surface area) than large conspecifics (Fig. 2). Corals with an ontogenetic decrease in SAP may experience an ontogenetic shift in growth limitation from biomass production to calcification. This geometry-based prediction is consistent with empirical measurements of tissue thickness for massive *Porites* which increases with colony size (*Barnes & Lough, 1992*). Based on geometry, microfragmention will not alter SAP for some colony morphologies such as branching corals with constant average branch diameter and foliose or encrusting corals with constant average thickness. The determining factor is whether the ratio of growth in surface area to growth in mass characteristically changes ontogenetically with size.

Corals with high aspect conical branches appear to have intermediate SAP values depending on their aspect ratio and average branch length. Their SAP values are between those of cylindrical branching corals and hemispheroidal corals with commensurate ecological implications. The aforementioned implications for ontogenetic shift in hemispheroidal coral can also apply to conical branching corals if their average branch length increases with colony size.

The coral morphology which naturally enables the highest SAP are thin plate-like shapes. In shallow water exposed to wave stress, sedimentation, and overshading competition, this fragile morphology is not common. However, in the lower photic zone, dominant photosymbiotic coral taxa (*e.g.*, *Leptoseris* spp. and *Agaricia* spp.) often mimics the shape of a flat circular discs (reviewed in *Kahng et al., 2019*; *Kramer et al., 2020*). At these depths offshore, hydrodynamic forces from storm events and exposure to sedimentation is often

attenuated. Space competition from fast growing photosynthetic taxa is also attenuated (reviewed in *Kahng et al., 2010*). Despite the very low calcification rates, the SAP for deep water *Leptoseris* spp. is higher than large, massive *Porites* spp. in shallow-water due to their hemispheroidal shape (*Kahng et al., 2023*).

In oligotrophic, shallow-water environments light available to drive daily photosynthesis is available from a wide range of angles due to scattering and the change in solar angle throughout the day (*Kahng et al., 2019*). However, in the lower photic zone, the angular width of usable light narrows centered around the vertical axis regardless of time of day. At these depths, planar area dictates the maximum amount of light that can be harvested by an organism. Therefore, the rate at which planar area is increased (*i.e.,* planar area productivity, PAP) becomes a strategic attribute for phototrophy. The flattening of coral morphology with depth (both interspecies differences and intra species plasticity) reflects an adaptation to increase PAP at depth. For example, the flattening of massive coral colonies with increasing depth (*e.g., Grigg, 2006*) increases both SAP and PAP. The coral colony morphology with a geometry that enables the highest PAP is a horizontally flat plate with a very thin skeletal thickness. This morphology is adopted by the deepest photosymbiotic corals around the world (*i.e., Leptoseris* spp.) and is consistent with phototrophic adaptation (*Kahng et al., 2012*).

### Funding
This study was funded by the Japan Science and Technology Agency (JST) Support for Pioneering Research Initiated by the Next Generation (SPRING) Grant Number PH8D230001. There was no additional external funding received for this study. The funders had no role in study design, data collection and analysis, decision to publish, or preparation of the manuscript.

### Grant Disclosures
The following grant information was disclosed by the authors:
Japan Science and Technology Agency (JST) Support for Pioneering Research Initiated by the Next Generation (SPRING): PH8D230001.

### Competing Interests
The authors declare there are no competing interests.

### Author Contributions

- Samuel E. Kahng conceived and designed the experiments, performed the experiments, analyzed the data, prepared figures and/or tables, authored or reviewed drafts of the article, and approved the final draft.
- Eric Odle conceived and designed the experiments, performed the experiments, analyzed the data, prepared figures and/or tables, and approved the final draft.
- Kevin C. Wakeman conceived and designed the experiments, authored or reviewed drafts of the article, and approved the final draft.

## Data Availability

The Coral Colony Geometry and Growth is available at GitHub and Zenodo:
- Available at https://github.com/ericodle/coral_geometry_modeling.
- EricOdle. (2024). ericodle/coral_geometry_modeling: publication release (v1.0). Zenodo. Available at https://doi.org/10.5281/zenodo.10668787.

## Supplemental Information

Supplemental information for this article can be found online at http://dx.doi.org/10.7717/peerj.17037#supplemental-information.

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
