# Peer review of "Coral geometry and why it matters"

_PeerJ, doi:10.7717/peerj.17037_

## Round 0.1 · original submission · Major Revisions

Two expert reviewers have evaluated your manuscript. As you can see from the comments below their overall evaluations are very different. However, both remark on the need to discuss the changes in geometry that coral colonies of all morphologies undergo over time and under different physiological conditions. I agree that this aspect should be included in your manuscript and encourage you to respond to the reviewers´ comments and make the appropriate changes in the manuscript.

Reviewer 1 ·

Basic reporting

I have some major concerns about this paper:

1. The authors introduce four geometric shapes to model several corals: hemispheroids, flat discs, branching corals with a constant branch parameter, conical branch/ colony
All four proposed equations are a very crude approximation of the real coral geometry. Can you find for example branching corals with a constant branch parameter? Even for relatively simple-shaped coral (e.g Montastrea annulearis) the hemispheroid approximation will not work. See for example the ``classical paper by Graus & Mcintyre , Variation in growth forms of the reef coral Montastrea annularis Ellis and Solander): a quantitative evaluation of growth response to light distribution using computer simulation, Smithson. Contr. Mar. Sci. 12:441-464, 1982. In the paper Graus McIntyre demonstrate that that the colony of one species can gradually transform from a hemi-spherical form , colum-shaped to plate-like shape depending on environmental conditions. Also in this example there no constant linear extension (line 240 in the manuscript) over the colony, but varies locally and depends on local light intensities on the colony. How can something like this ever be captured by the four basic geometries in this paper. I doubt if there is any coral with constant extension rate over the surface, the local extension rate is always related to the local physical environment.

2. Currently there is large number of papers available of authors who have tried to quantify volumes, surfaces (and many other quantitative properties!) using 3D laser scans, CT scans etc. in great detail The authors seem to be barely aware of these approaches (they only briefly mention this approach in lines 272-276) in the discussion. What is the advantage of their approach compared these much more realistic 3D measurements?


Unfortunately I cannot recommend this paper for this publication.

Experimental design

see above (major concerns)

Validity of the findings

see above (major concerns)

Cite this review as

·

Basic reporting

Some minor English editing is needed.

The article includes sufficient background that is appropriately referenced.

Tables and figures are relevant to the content. Nevertheless, there is room to improve figure resolution. Additionally, some labels should be corrected.

The article is self-contained with relevant results to hypotheses.

Experimental design

Original and within the aims and scope of the journal.

Research is well-defined, relevant, and meaningful.

Within the research boundaries (within a small coral geometry), the investigation was conducted rigorously.

Methods described with sufficient detail.

Validity of the findings

The article provides sound results with implications for coral physiology and ecology within a relatively small spectrum of coral geometries. However, the manuscript will benefit from a discussion of the changes in geometry that corals experience as they grow and how this relates to their results and research. For instance, most juvenile or small corals, irrespective of adult shape (i.e. not restricted to hemispheric shapes), have significant growth rates and successful microfragmentation when they are small.

See comments in the attached manuscript.

Cite this review as

---

## Round 0.2 · accepted · Accept

Two experts have re-evaluated your manuscript. As you can see one reviewer recommends acceptance and the other recommends rejection due to the limitations of simple geometric models. However, after having read your manuscript, I see that you have indeed included text in the manuscript about the limitations of using such models and include examples of where these models are useful and context for when they should not be used. Therefore, I am recommending that this manuscript be accepted for publication.

Reviewer 1 ·

Basic reporting

I still have major concerns about this paper. In the response the authors claim that the simple geometrical models have a high accuracy. This may true be that when comparing the simple geometric models with 3D photogrammetry and 3D computer tomography (CT) scans (Courtney et al. 2007; Naumann et al. 2009) that in some aspects there is a high correspondence. The problem remains that the simple geometric models are only descriptive and do not capture the full complexity of the coral morphology. It will be never possible to correctly capture quantities like the coral surface, rugosity, branch spacing, surface curvature and all kind of other 3D quantities which can be measured in great detail in 3D scans. For a better understanding of all kind biochemical processes quantities as for example the surface of the coral exposed to the environment and estimation of boundary layers, these measurements are crucial.

I don’t agree with the response of the authors that the observations in the coral species (Montastrea annularis) from Graus and McIntyre (1982) are some exception supporting an overgeneralization. I think many coral species are characterized by a strong morphological plasticity as a response to the environment (e.g T. Nakamura and R. van Woesik, water-flow rates and passive diffusion partially explain differential survival of corals during the 1998 bleaching event, Mar. Ecol. Prog. Ser., 2001, 212:301-304). This morphological response is important for the survival of coral species.

In summary I think that the simple geometric models are a fully descriptive method and can never be used to obtain a deep understanding of relevant biochemical processes (e.g the calcification process) and environmental influence (environmental DIC, pH) (. I think these limitations of the simple geometric models should clearly be described. May it is possible to mention applications where these simple geometric models might be useful.

Experimental design

see above

Validity of the findings

see above

Cite this review as

·

Basic reporting

No further comment

Experimental design

No further comment

Validity of the findings

No further comment

Additional comments

In its actual form, the manuscript is ready for publication.

Cite this review as